# SUPER and femtosecond spin-conserving coherent excitation of a tin-vacancy color center in diamond

Cem Güney Torun [1,6], Mustafa Gökçe[1,6], Thomas K. Bracht[2], Mariano Isaza Monsalve[1,3], Sarah Benbouabdellah[1], Özgün Ozan Nacitarhan[1], Marco E. Stucki [1,4], Domenica Bermeo Alvaro [1,4], Matthew L. Markham [5], Tommaso Pregnolato [1,4], Joseph H. D. Munns[1], Gregor Pieplow [1], Doris E. Reiter [2] & Tim Schröder [1,4] ✉

The coherent excitation of an optically active spin system is one of the key elements in the engineering of a spin-photon interface. Using the novel SUPER scheme, we coherently control the main optical transition of a tin-vacancy color center in diamond with nonresonant ultrashort optical pulses. Furthermore, we implement a femtosecond control scheme using resonant pulses for achieving record short quantum gates applied to diamond color centers. We simulate the applicability of the SUPER scheme to spin qubits and experimentally investigate spin mixing. Finally, we propose a spin-spin entanglement scheme in a scenario where the excitation with broadband pulses is incompatible with spin-selective excitation. The employed ultrafast quantum gates open up a new regime of quantum control with solid-state color centers, enabling multi-gate operations and efficient spectral filtering of the excitation laser from deterministically prepared coherent photons.

The deterministic and coherent creation of single photons is a core element of a variety of quantum technology applications such as quantum networking[1] and photonic quantum computation[2]. To this end, the negatively charged tin-vacancy color center (SnV) (Fig. 1a) has attracted increasing attention in recent years. This interest stems from its desirable qualities, such as first-order resistance to spectral diffusion[3–5] and extended coherence times at elevated temperatures compared to other group-IV color centers in diamond[6]. Using these properties, a plethora of applications with the SnV have been demonstrated, such as quantum memories[7,8], and coherent single-photon generation[9].

Despite the recent progress, the creation of spin-photon or remote spin-spin entanglement with the SnV has so far been elusive. A major challenge is the creation of single photons entangled with the SnV's electron spin degree of freedom, which requires optical

excitation that maintains coherence of the emitted photons. Although resonant excitation provides an effective means to coherently prepare excited states, it comes with a fundamental drawback: filtering the excitation light from the subsequently emitted single photons. Since both modes will have the same carrier frequency, it is not possible to use spectral filtering to separate them. Instead, mode separation methods between the excitation and detection field can be applied in the polarization, time or spatial domain. While the latter is well established for atomic systems with high extinction[10], any of the mode separation methods are challenging for solid-state quantum systems.

The most established method to circumvent this issue is using cross-polarization filtering[11,12]. This method, however, eliminates at least half of the photons emitted by the quantum system. Temporal filtering of the excitation light equally introduces losses[13]. Spatial mode

[1]Department of Physics, Humboldt-Universität zu Berlin, Berlin, Germany. [2]Condensed Matter Theory, TU Dortmund University, Dortmund, Germany. [3]Institute for Experimental Physics, Universität Innsbruck, Innsbruck, Austria. [4]Ferdinand-Braun-Institut (FBH), Berlin, Germany. [5]Element Six, Harwell, UK. [6]These authors contributed equally: Cem Güney Torun, Mustafa Gökçe. ✉e-mail: tim.schroeder@physik.hu-berlin.de

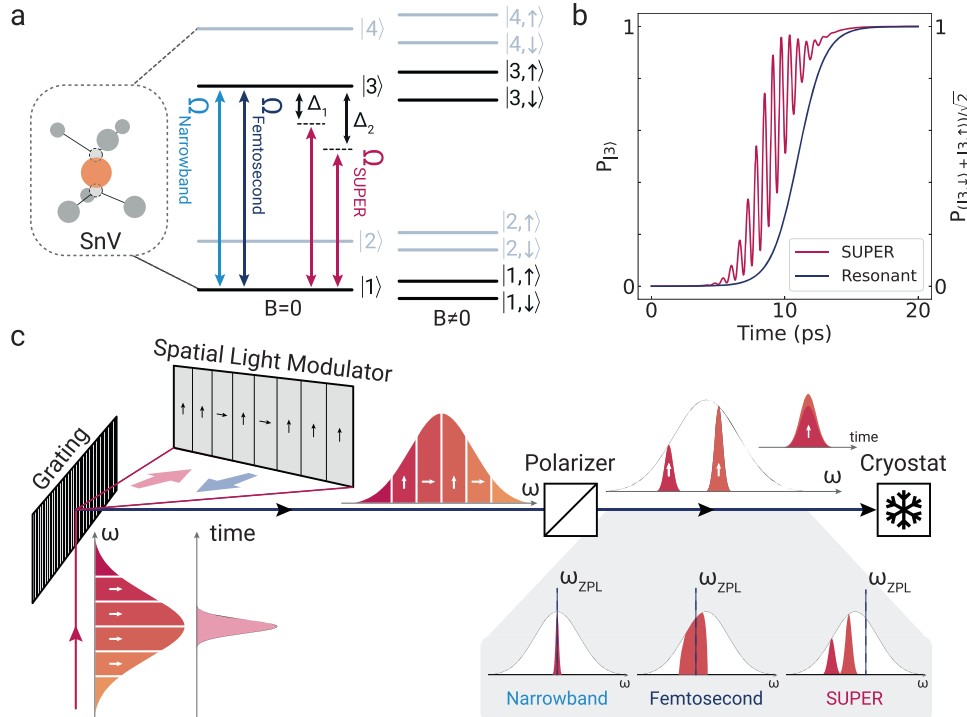

**Fig. 1 | The tin-vacancy color center and its optical control. a** Energy level manifold of the SnV with and without an applied magnetic field. **b** Example of population inversion dynamics from ground to excited states, for orbital ($|1\rangle \rightarrow |3\rangle$) and spin states ($\frac{1}{\sqrt{2}}(|1, \downarrow\rangle + |1, \uparrow\rangle) \rightarrow \frac{1}{\sqrt{2}}(|3, \downarrow\rangle + |3, \uparrow\rangle)$), using resonant or SUPER schemes. **c** Simplified experimental pulse carving setup. The spectral bands of a broadband pulse are dispersed spatially with the help of a diffraction grating and reflected onto a spatial light modulator (SLM). By defining slits on the pixels of the SLM, it is possible to change the polarization of the desired frequency bands.

Undesired frequencies are filtered out at the output with the help of a polarizer. Using the pulse carver it is possible to engineer pulses with different spectral shapes from a broad bandwidth Gaussian pulse. In this work, we use three configurations: (i) a resonant single *narrowband* pulse, (ii) resonant semi-broadband quadrilateral-like *femtosecond* pulse, (iii) nonresonant two-color detuned Gaussian narrowband pulse to implement the *SUPER* scheme. The real pulse spectra employed in the experiment are presented in the Supplementary Fig. 3.

separation of the excitation and collection modes is yet another also not simple solution. It requires complex photonic structures such as waveguides or resonators to make the single emitters couple efficiently to well defined spatial modes[14], primarily in the collection path.

Another possibility of eliminating laser fields for coherent excitation are off-resonant excitation schemes that allow for the separation of excitation and signal using simple spectral filters. While the previously discussed polarization, time or spatial domain mode separation methods allow for using resonant excitation, a process well understood and explored in quantum optics, off-resonant coherent schemes in contrast rely on far less-explored light-matter interaction mechanisms. The first ideas go back to the continuous wave (CW) off-resonant excitation of dressed two-level systems[15–18]. Going from the CW to the pulsed regime in off-resonant driving, as required, for example, in entanglement protocols, is a relatively recent area of research. Few proposals and demonstrations have so-far been realized: notch-filtered adiabatic rapid passage[19], dichromatic, phase-locked excitation[20,21], and the Swing-UP of the quantum EmitteR population (SUPER) scheme[22–24]. These works, however, focus on two level systems only and do not consider spin degrees of freedom as required for spin-photon or spin-spin entanglement generation. Moreover, all previous works have been realized with optically driven semiconductor quantum dots, which have excellent optical but poor electron spin properties[25].

In this work, we demonstrate the non-resonant coherent excitation of an optically active spin defect in diamond that has shown millisecond coherence times[26]. We use the SUPER scheme, which was previously introduced by some of the authors[27]. The SUPER scheme relies on two red detuned excitation pulses, that under particular detuning and pulse power configurations can coherently invert the spin defect's optical population (Fig. 1b). Due to the hundreds of GHz detuning, spectral filtering is suitable with this method. By extending the original model with the spin manifold of group-IV defects in diamond we show theoretically and demonstrate experimentally that SUPER control pulses are compatible with spin preparation, control, and read-out, a quintessential requirement for creating spin-photon entanglement. Furthermore, we study the influence of the applied picosecond pulses on the spin properties of the SnV and demonstrate that the spin sub-levels are not affected by the optical fields. To possibly extend the SUPER regime towards ultra-fast pulses in the future, we explore resonant coherent control with femtosecond optical pulses and demonstrate optical Rabi oscillations with GHz rates.

Following our experimental demonstration, we introduce a protocol for a spin-spin entangling scheme that addresses the unique properties of our control pulses. Using ultrashort and therefore spectrally broad pulses entails a specific challenge for creating entanglement due to the prohibited spin selective excitation[28]. Therefore, we develop a protocol where both spin transitions are simultaneously excited and the emitted photons are encoded in the frequency basis.

## Results

### Pulse carver

The core device for enabling the introduced and even further coherent control schemes is the pulse carver (Fig. 1c), which is a modified version of a commercial pulse slicer (APE f50) that allows us to convert a

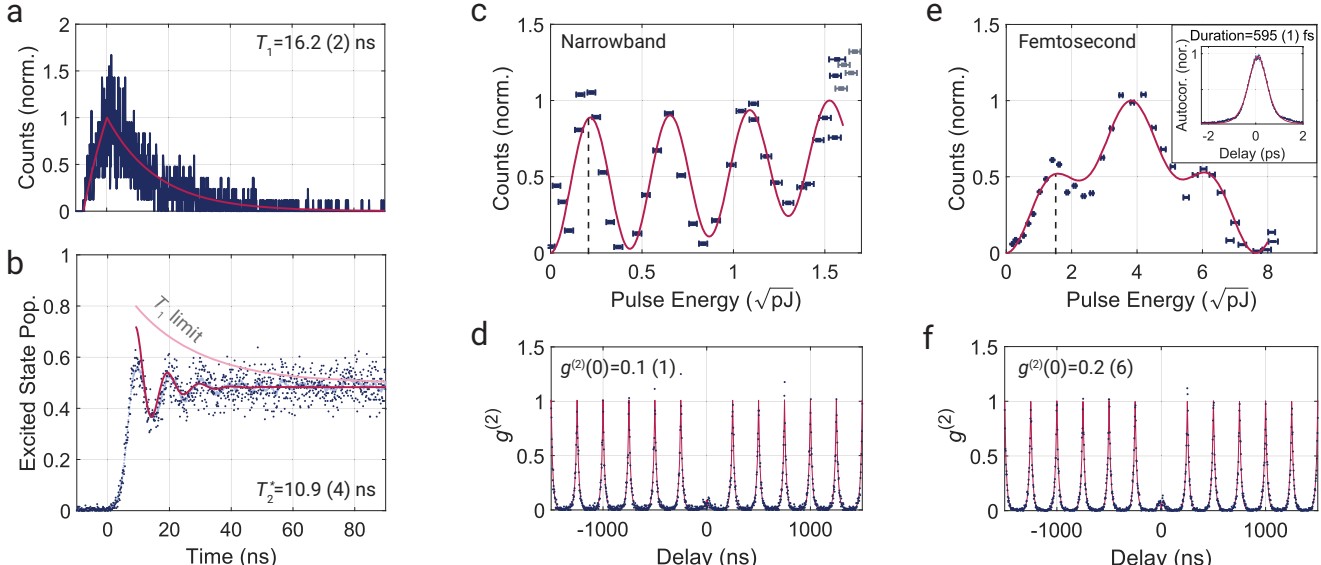

**Fig. 2 | Coherence measurements and quantum control of the SnV. a** Lifetime measurement of the SnV. By applying the narrowband pulse at $\pi$-rotation power and analyzing the fluorescence response, we extract an average spontaneous emission lifetime of 16.2 (6) ns, which corresponds to the $T_1$ of our optical qubit. **b** Time-resolved Rabi oscillations under a quasi-continuous excitation scheme, where a continuous wave laser is switched on with an acousto-optic modulator, and the fluorescence response is recorded. The observed Rabi oscillations establish the coherence of the investigated qubit. From the amplitude decay of the oscillations, we estimate an overall dephasing of the optical qubit under the experimental conditions as $T_2^* = 10.9$ (4) ns. **c** Optical Rabi rotations extracted by varying the power and integrating the total amount of registered photons using a ~40 GHz wide pulse of 15 ps duration. The dashed line indicates a $\pi$-rotation. **d** Photon purity measurement of the emitter fluorescence at a power calibrated from (**c**, dashed line) to perform a $\pi$-rotation. The estimated $g^{(2)}(0)$ is 0.1 (1), which confirms the

single-photon nature of the collected emission. **e** Optical Rabi rotations extracted by varying the pulse power and integrating the total amount of registered photons using a 1400 GHz wide quadrilateral broadband pulse with femtosecond duration. The dashed line indicates a $\pi$-rotation. The peak around 4 $\sqrt{pJ}$ corresponds to the $3\pi$ rotation. Inset: Measurement of the pulse duration using an autocorrelator. Full-width-half-maximum of the $sech^2$ autocorrelation signal is multiplied by a factor of 0.647 to deconvolve the pulse duration. **f** Photon purity measurement of the emitter fluorescence at a power calibrated from (**e**, dashed line) to perform a $\pi$-rotation. With the estimated $g^{(2)}(0) = 0.2$ (6), the single-photon emission is confirmed but a higher background contribution compared to narrowband pulses is detected. Horizontal error bars in c and d correspond to the error specified by the powermeter sensor head. The remaining uncertainties in the figure are extracted from the 95% confidence intervals returned by the fit algorithms.

spectrally wide (i.e., temporally short) pulse into a pulse with an arbitrarily shaped spectrum within the original pulse shape. We use this approach to generate narrowband picosecond pulses, broadband femtosecond pulses, as well as pulses containing two spectral bands for the SUPER scheme. To do so, a ~150 fs pulse with a bandwidth of 3.6 nm is spatially dispersed via a reflective diffraction grating. A cylindrical lens focuses individual frequency bands onto the pixels of a spatial light modulator (SLM). We can set the voltages of individual pixels such that the polarization of selected frequency bands are altered. The altered frequency bands are subsequently filtered out using a polarizer, resulting in a pulse with the desired spectral shape and time-bandwidth.

The shaped pulses are then directed via a dichroic mirror into a confocal microscope, which is coupled into a helium cryostat operating at 4.5 K. The cryostat cools the negatively charged tin-vacancy color centers in diamond (SnV) embedded into diamond nanopillars down to the operation temperature. The nanopillars enhance the photon collection efficiency compared to bulk[29]. After the excitation, the control pulses are spectrally filtered, and photons from the phonon sideband are collected.

### Optical qubit coherence

In the absence of a magnetic field, the SnV has four distinct energy levels (Fig. 1a), two in the ground state $|1\rangle$ and $|2\rangle$, and two in the excited state manifold $|3\rangle$ and $|4\rangle$. All levels are two-fold spin degenerate, and experience Zeeman splitting when a magnetic field is present[30]. Here, we focus on the optical transitions between levels $|1\rangle$ and $|3\rangle$, conventionally named C transition. The spin states associated with the long-lived ground state $|1\rangle$ are the most viable for use as a

qubit[6,31,32]. Therefore, the C transition is the most desirable to control for extracting coherent photons.

The optical coherence of a transition is of great importance when devising an emitter-photon interface. We therefore first quantify the coherence of the optical qubit composed of the states $|1\rangle$ and $|3\rangle$ under zero magnetic field.

We start by determining the lifetime of the optical qubit $T_1$, which corresponds to the inverse of the spontaneous emission rate. It can be obtained by analyzing the fluorescence response after being excited by a pulse much shorter than $T_1$. Triggered by the pulse, we record the time it takes for a photon to be registered on an avalanche photodiode after excitation. From the arrival statistics, we find the exponential decay of the excited state population (Fig. 2a). After deconvolving the data and instrument response time function, we extract a lifetime of $T_1 = 16.2$ (6) ns, approximately two times longer than previously measured for SnVs[33–35]. The lifetime constitutes an upper limit for the coherence of the optical qubit. We attribute the discrepancy between our results and the values reported in the literature to two main factors. First, the emitter is embedded in a nanopillar, where the effective refractive index is lower than that of the bulk material and the local density of optical states is reduced[36]. Consequently, the spontaneous emission rate is expected to be suppressed. Second, in our confocal microscope configuration, the excitation and collection paths overlap at a polarization beam splitter. The excitation mode is aligned with the transition dipole of the C transition of the SnV center, resulting in the collection of light with the orthogonal polarization. The collected signal predominantly consists of photons emitted via the D transition, thereby restricting the detected emission to a single optical transition and effectively increasing the lifetime.

The second figure of merit we investigate is the optical dephasing $T_2^*$ time. A quasi-continuous resonant driving of the C transition of the SnV is used for this measurement. Using a continuous wave (CW) resonant laser (CW power: 1.5 μW), and utilizing an acousto-optic modulator to switch the laser on demand, we drive the qubit and record the arrival of the photons with respect to the starting time of the laser pulse. Fig. 2b plots the histogram of the detected photons as a function of time. A temporal oscillation of the fluorescence signal is clearly visible. These oscillations can be interpreted as coherent Rabi oscillations between levels $|1\rangle$ and $|3\rangle$.

We estimate an overall optical dephasing from the decay of the Rabi rotations by fitting the data with the time evolution of a two-level system in the presence of spontaneous decay of the excited state and pure dephasing (see Supplementary Section II B) and extract $T_2^* = 10.9$ (4) ns. We note that we cannot distinguish individual dephasing mechanisms, such as intrinsic dephasing, spectral diffusion, and laser-induced dephasing separately and can only provide an overall rate. These distinctions can be extracted using a Ramsey method in combination with a linewidth measurement[9,37]. Although the dephasing time depends on the experimental details such as the applied laser's power and phase stability, we indeed observe a partially coherent evolution of the optical qubit. Further characterization measurements on the investigated emitter can be found in the Supplementary Fig. 6, including spectral data verifying emission from an SnV.

## Ultrafast resonant coherent control

Switching the field off at the desired state during resonant driving of a two-level system, i.e., generating a pulse with a controlled pulse area, allows the preparation of a desired state. Quantum gates can therefore be implemented by tuning the pulse duration or power[38,39]. So far, spin-photon entanglement has traditionally been created by carving optical pulses with electro-optical modulators. Such pulses generally have a duration of 1–2 ns and result in spin-photon fidelities ranging from 77%[9] to 95%[40]. Here, we use ultrashort pulses from a mode-locked laser[41] to realize pico- and even femtosecond quantum control.

For the fast pulsed Rabi control experiment, we use pulses with 42.3 (2) GHz spectral width. This bandwidth corresponds to the narrowest pulse that can be generated within our spectral SLM-based filter and at the same time allows for a high signal-to-background ratio. We estimate a pulse duration of ~15 ps using an autocorrelator connected to a diagnostic path accessed with a flip mirror placed before the confocal microscope. The measured pulse duration is about 50% longer than 10.4 ps, the duration expected from a Gaussian time-bandwidth product limited pulse. The reason for the discrepancy is the dispersion introduced in the setup. Details of the autocorrelation measurements on the generated pulses are provided in Supplementary Section I B 1.

The observed oscillations in the fluorescence response of the emitter during a variation of the pulse's energy constitute evidence for coherent control. We demonstrate optical Rabi rotations up to $7\pi$ (Fig. 2c). For fitting the data, we employ a function based on a decaying oscillation and a background term: $A\sin^2(\omega\sqrt{P_e})\exp(\gamma P_e) + CP_e$, where $A$ (amplitude), $\omega$ (modulation), $\gamma$ (decay rate) and $C$ (background coefficient) are the fit parameters, and $P_e$ is the pulse energy. We exclude higher energy data points from the fit because of the sudden and drastic change of the system response, possibly caused by a threshold-based nonlinear effect.

A direct estimation of the gate fidelities for a $\pi$-pulse is not possible since there is no available method to probe the absolute excited state population. Instead, in Supplementary Section II C we rely on an indirect method and fit the first ~$4\pi$ oscillations to the same model based on a damped oscillator with a linear background contribution.

Finally, we measure the photon purity, which is an additional figure of merit that characterizes the emission. We implement a Hanbury Brown-Twiss interferometry experiment at $\pi$-pulse power and find $g^{(2)}(0) = 0.1$ (1) (Fig. 2d), which shows clear antibunching and single photon emission from the ultrashort pulse driven emitter.

Next, we investigate the shortest pulse duration which can still coherently excite the emitter. Using an autocorrelation measurement, we identify pulses with femtosecond duration (Fig. 2e, inset). In the femtosecond regime, the pulses have a bandwidth of ~1400 GHz. The spectra of the wide bandwidth pulses are no longer Gaussian, but rather present a quadrilateral shape, due to the way they are sliced with the SLM. Since the closest optical transition is about 820 GHz detuned, the selected bandwidth is at the limit for avoiding resonant cross-coupling between levels $|2\rangle$ and $|3\rangle$ (820-1400/2 = 120 GHz). By varying the pulse amplitude of the broadband pulses, we achieve a $6\pi$ rotation as shown in Fig. 2e. We fit the data to a superimposed bi-sinusoidal model to better capture the phenomenological trends observed in the data: $A_1\sin^2(\omega_1\sqrt{P_e}) + A_2\sin^2(\omega_2\sqrt{P_e})$ where $A_{1,2}$ (amplitudes) and $\omega_{1,2}$ (modulation) are the fit parameters. The data shows nonperfect fluorescence extinction of the Rabi rotations at $2\pi$ and $4\pi$, and a high inversion at $3\pi$ pulse energies. The observed behavior may result from variations in cross-coupling to other electronic levels of the SnV during the control pulse or from nonlinear effects induced by high excitation energies. The increased pulse energy requirements for femtosecond control in comparison with narrowband (e.g., ~40 times higher for a $\pi$-pulse) results in an increased background, which in turn increases $g^2(0)$ to 0.2 (6) (Fig. 2f) at the $\pi$-pulse power.

Enabling coherent operations on this time scale is especially beneficial for analyzing ultrafast processes induced by light-matter interaction[42]. Fast coherent optical control could also be relevant in the presence of large Purcell enhancement, which reduces the lifetime of emitters significantly and therefore makes faster gates a necessity, particularly for generating single photons in high repetition or for the creation of cluster states[43]. Ultrashort pulses can also enable applications requiring multiple gate operations on the optical qubit in a limited time window[44]. For example, a Hahn-echo type, phase-sensitive measurement, could extend the SnVs' sensing capabilities[45].

## Nonresonant swing-up control: SUPER

The SUPER scheme achieves coherent excitation by relying on a highly red detuned pulse pair with different central frequencies, which by themselves cannot significantly transfer the population to the excited state[27]. The pulse configuration provides an advantage compared to the resonant scheme because the photons from the laser pulses can be spectrally distinguished from the signal. This benefit makes separation of excitation light and the emitted photons much more feasible, while maintaining their purity and indistinguishability. For finite pulses, the achievable final occupation can be optimized by fine-tuning pulse parameters such as amplitude, detuning, and bandwidth.

Although the SUPER scheme can work seamlessly with any two-level system with coherent transitions and lifetimes significantly longer than the applied pulses' durations, it must be noted that most color centers have more complicated electronic structures. Setting the pulse parameters, therefore requires careful alignment of the spectral positions of the pulses such that they do not couple to other transitions. Furthermore, the resonance widths of the SUPER excitation are determined by the bandwidths of the two-color pulses, and do not facilitate individual control of multiple emitters in a single diffraction-limited spot if the central frequencies are contained within the resonance.

The resonance condition of the SUPER scheme is determined by the pulse with the smaller detuning. When the parameters for the less detuned pulse are varied, the conditions set upon the more detuned pulse also change[46]. Experimentally, we find the right configuration of pulses by fixing one pulse at a set frequency, bandwidth, and amplitude; and then performing a scan over detuning and amplitude of the second pulse. Our fixed pulse has a pulse area that would result in a $7\pi$ rotation if applied resonantly, and a 116.6 (2) GHz detuning from the

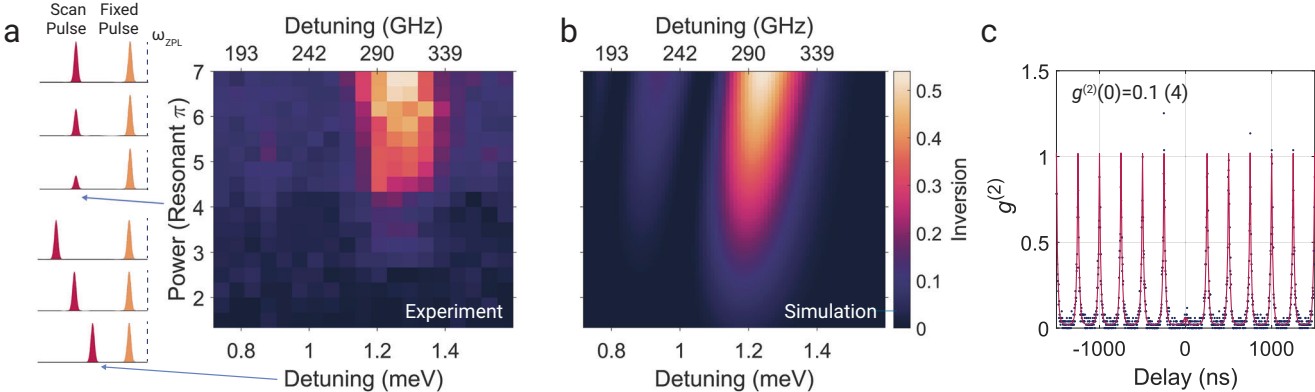

**Fig. 3 | Implementation of the SUPER scheme. a** Experimental data obtained by sending two-color nonresonant pulses while one of the two pulses' (the scan pulse) power and detuning to the SnV resonance is varied. The pulse power of the fixed pulse is set to maximum available (corresponding to ~7π), and its detuning to the SnV resonance is set to 117 GHz. The inversion is calculated by taking the ratio of the absolute number of registered counts after applying the two-color SUPER pulse and a resonant narrowband π-pulse. **b** Theoretical simulation of the SUPER excitation using the experimental pulse parameters. A good agreement between the experimental data and simulation is observed. **c** Photon purity measurement conducted to the emission at the maximum emission. $g^{(2)}(0)$ of 0.1 (4) is obtained. This implies a single-photon emission from an individual emitter. Uncertainty is extracted from the 95% confidence interval retrieved from the fit algorithm.

central frequency of the emission. After sweeping the second pulse's amplitude and detuning, we observe fluorescence occurring in the region of 308 (11) GHz detuning at our maximum achievable power (Fig. 3a). We estimate the maximum inversion by comparing the counts with a resonant π-pulse and estimate 55 (1)%. For verification, we apply the same measurement with either one of the pulses in isolation. In both cases, a negligible emission is observed (see Supplementary Fig. 10). We can therefore confirm that the emission indeed comes from the collective action of both pulses.

We compare the measured fluorescence with a simulation of the coherent excitation using a theoretical model conducted in the experimental scan range and respective pulse parameters. The simulation results are shown in Fig. 3b. They show excellent quantitative agreement, in terms of the position of the resonance at the maximal power (experiment: 308 (11) GHz, simulation: 298 GHz) and inversion (experiment: 55 (1)%, simulation: 54%).

We also conduct an autocorrelation measurement of the emission to estimate the photon purity (Fig. 3c). We obtain $g^{(2)}(0) = 0.1$ (4) providing convincing evidence for single-photon emission.

Both the experimental and theoretical results confirm that with our current setup, we can achieve a π/2 rotation with the generated SUPER pulses. Since the non-unity inversion fidelity is a result of the limited experimental pulse powers, we extend the simulation to higher powers. We find that it is theoretically possible to increase the inversion fidelity up to 99.8% by simply increasing the power of the laser pulses (see Supplementary Fig. 11). Therefore, we conclude the implemented method is capable of achieving full inversion and facilitating deterministic photon extraction.

We would also like to point out that the investigated emitter in this and the previous section was only preselected on the condition that it was a single emitter embedded in a nanopillar, and no further effort was required to find an emitter particularly suitable for demonstrating the SUPER scheme.

## Spin state properties under SUPER control

So far, we have presented results for an excitation scheme at zero magnetic field strength, which does not provide access to the coherent and long lived spin qubit levels[7,26,47], essential for quantum applications requiring spin-photon or spin-spin entanglement[48]. For this purpose, we simulate the coherent inversion of the ground spin states to the excited states. We perform the simulations using the full SnV Hamiltonian, including the Zeeman interaction. The details of the SnV Hamiltonian and the interaction with the driving field can be found in ref. [49]. The initial state is the equal superposition $|\Psi_0\rangle = \frac{1}{\sqrt{2}}(|1, \downarrow\rangle + |1, \uparrow\rangle)$ (see Fig. 1a). We then optimize the SUPER pulses for a magnetic field oriented at the SnV high-symmetry axis for the target state of $|\Psi\rangle = \frac{1}{\sqrt{2}}(|3, \downarrow\rangle + |3, \uparrow\rangle)$. We show that 99.8% inversion fidelity is theoretically possible using the SUPER scheme (Fig. 4a, b). Spontaneous emission results in the entangled spin-photon Bell state $|\Psi\rangle = \frac{1}{\sqrt{2}}(|1, \downarrow\rangle|1_{\omega_\downarrow}\rangle + |1, \uparrow\rangle|1_{\omega_\uparrow}\rangle)$.

In addition to the symmetric B-field alignment, we also simulate SUPER pulses with arbitrarily oriented fields and find that coherent population exchange can occur between spin levels. This can be counteracted by preparing an unequal superposition prior to the excitation resulting in the desired equal superposition. A discussion of an alternative excitation scheme involving broadband excitation pulses can be found in the Supplementary Section IV B.

Next, we experimentally probe the induced spin mixing after applying SUPER pulses to an emitter embedded in a different nanopillar in the same sample. We conduct a spin state lifetime measurement implemented in three steps (Fig. 4c). First, we apply a resonant spin state initialization pulse (CW power: 1 μW) and pump the population from one spin level to the other using weak spin flipping transitions. A variable delay then allows for thermalization to occur between the spin levels. We conduct two measurements: one with the SUPER pulses applied after the spin state initialization and one without. Finally, we apply a readout pulse. From the fluorescence signal, we determine how much of the population has thermalized, returning the system to the initial state. Due to the phononic processes dominant at 4.5 K[7], the initialization fidelity is only ~70%. We fit the population recovery with an exponential decay $P_\sigma(t) = P_0 \exp(-t/T_{1,\text{spin}})$ and extract the time constants $T_{1,\text{spin}}$ of the spin state. We find $T_{1,\text{spin}}^{\text{SUPER}} = 47$ (14) μs with the SUPER pulses (Fig. 4d), and $T_{1,\text{spin}}^{\text{control}} = 41$ (11) μs without (Fig. 4e). Having matching values within fit uncertainties is consistent with the SUPER pulses not inducing observable spin mixing. We attribute the small variation of the $T_1$ times to the slow wandering of the emitter resonance fluctuating initialization fidelities and non-aligned magnetic field orientation.

Finally, we discuss how the SUPER scheme can be utilized when employing a color center as a spin-photon interface for linking remote quantum nodes. Such protocols typically rely on spin-selective excitations and time-bin encoding of photons, which are, however, not feasible due to the broadband nature of the SUPER scheme. Therefore,

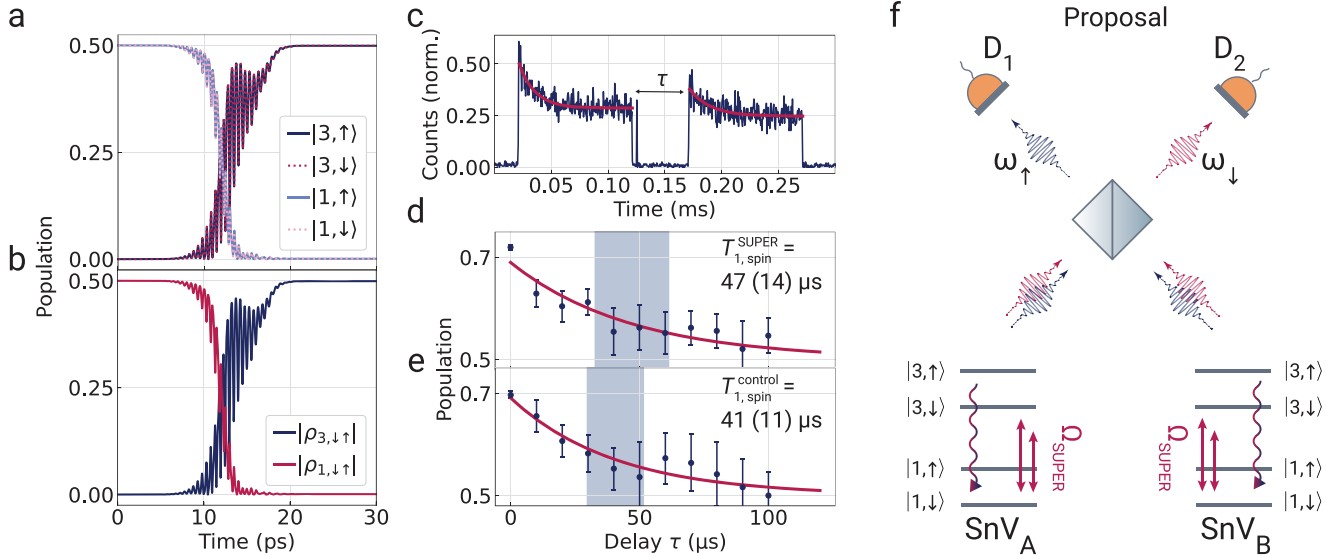

**Fig. 4 | SUPER applied to spin qubits. a** Simulation of the temporal evolution of the ground and excited states of the spin qubit under an optimized magnetic field and SUPER pulses. Population exchange dynamics of $|1, \downarrow\rangle \to |3, \downarrow\rangle$ and $|1, \uparrow\rangle \to |3, \uparrow\rangle$ follow a similar trajectory. **b** Coherence transfer from the ground state superposition, to the excited state superposition: $|\rho_1, \downarrow\uparrow| = |\langle 1, \downarrow |\hat{\rho}| 1, \uparrow\rangle| \to |\rho_3, \downarrow\uparrow| = |\langle 3, \downarrow |\hat{\rho}| 3, \uparrow\rangle|$. **c** Subsequentially applied spin initialization and readout pulses with a variable delay $\tau$ for estimating spin coherence. The observable peak at ~ 0.13 ms is the fluorescence induced by the applied SUPER excitation. **d** The thermalization of the spin states when a SUPER pulse is applied. Decay time of the population to the mixed state yields the $T_{1,\mathrm{spin}}$ time. **e** The control measurement for spin population thermalization when no SUPER pulses are applied between initialization and readout pulses. Uncertainties are extracted from the 95% confidence intervals retrieved from the fit algorithm. **f** Illustration of the proposed entanglement scheme based on generating photonic qubits encoded in the frequency basis. Two SnVs are first prepared in an equal spin superposition in state. Next, using the SUPER scheme population is inverted to excited state, while maintaining the probability amplitudes of the spin levels. Finally, the two SnVs emit photons with frequencies entangled to their spin-state. Using Hong-Ou-Mandel interference, detecting two clicks on two detectors heralds the state when two photons are distinguishable and the spin states of two SnVs are anti-correlated.

inspired by the Barret-Kok scheme[13,28], we propose a heralded spin-spin entanglement scheme using the SUPER approach for generating spin-entangled states between distant nodes (Fig. 4f): First, prepare equal superpositions of two spin systems labeled $\sigma$ = A, B such that their initial states become $\frac{1}{\sqrt{2}}(|1, \downarrow\rangle_\sigma + |1, \uparrow\rangle_\sigma)$. After the SUPER excitation $\frac{1}{\sqrt{2}}(|3, \downarrow\rangle_\sigma + |3, \uparrow\rangle_\sigma)$ is created, which, due to spontaneous emission ideally evolves into $\frac{1}{\sqrt{2}}(|1, \downarrow\rangle_\sigma |1_{\omega_\downarrow}\rangle + |1, \uparrow\rangle_\sigma |1_{\omega_\uparrow}\rangle)$.

If the emitted photons are superposed on a beam splitter, the anti-correlated Bell state $|\psi_-\rangle = \frac{1}{\sqrt{2}}(|1, \downarrow\rangle_A |1, \uparrow\rangle_B - |1, \uparrow\rangle_A |1, \downarrow\rangle_B)$ is heralded if both detectors click (see Supplementary Section V for details). Interestingly, the heralding step relies on the non-interfering, distinguishable photons. The fact that both detectors have to click also makes this scheme sensitive to photon loss. A $\pi$-rotation of the spin states after the successful first heralding step and a repetition of the heralding protocol would remove any additional unknown phase[13].

## Discussion

We have implemented the SUPER scheme using two detuned pulses with a color center in diamond and demonstrated nonresonant optical $\pi/2$ gates. By implementing this method, we introduce less demanding spectral filtering methods for isolating the zero-phonon line emission of color centers in diamond. Extending the application of the SUPER scheme to a new quantum platform after semiconductor quantum dots also solidifies its suitability with emitters in solid state materials like silicon carbide and silicon, as well as atomic systems.

Furthermore, we demonstrate ultrafast quantum control of an SnV color center, a core step for further developing small time bandwidth building blocks of quantum information processing. We also reduce the resonant control scheme pulse length under one picosecond, a record time scale for color centers in diamond, paving the way for multi-gate coherent control of emitters in resonators with short lifetimes.

The demonstrated coherent control schemes of the optical qubit can be integrated with spin qubit control of the SnV. We have shown theoretically that it is possible to fully excite a spin qubit using the SUPER scheme and experimentally collected evidence that such a scheme would not induce spin-mixing. The proposed spin-spin entanglement scheme paves the way for using the SUPER scheme on multi-qubit entanglement schemes.

## Methods

### Sample

The sample used in this work is a single-crystal electronic grade diamond grown by chemical vapor deposition (supplied by Element Six Technologies Ltd. (UK) and with {100} faces). The substrate is initially cleaned for about 1 h in a boiling triacid solution ($H_2SO_4$:$HNO_3$:$HClO_4$, 1:1:1)[50] and then etched in $Cl_2$/He and $O_2$/$CF_4$ plasmas, in order to remove any organic contaminants and structural defects from the surface[51]. Sulfur ions are initially implanted into the substrate, with a nominal fluence and implantation energy of $5 \times 10^{12}$ atoms $cm^{-2}$ and 160 keV, respectively. A low pressure-high temperature (LPHT) annealing step (temperature $T = 1050\,°C$ and $P \approx 1 \times 10^{-7}$ mbar), for about 12 h, is then used to heal the lattice damage caused by implantation process. Sulfur implantation step is included for enhancing the final emitter yield[52], but has not yet been characterized on this sample. After a second triacid cleaning step, tin ions are implanted in the same substrate, with nominal values $5 \times 10^{10}$ atoms $cm^{-2}$ and 400keV for fluence and energy, respectively. According to Stopping and Range of Ions in Matter[53] simulations, the expected implantation depth is about 100 nm for both S and Sn. After a second LPHT annealing step, the substrate is annealed for the third time at 2100 °C at $P \approx 6$ GPa for a total of 2 h and finally cleaned in a boiling triacid solution.

Nanopillars are fabricated by e-beam lithography and plasma etching. First, a 200 nm-thick layer of $SiN_x$ is deposited on the surface

of the diamond in an inductively coupled-plasma (ICP) enhanced chemical vapor deposition system. After spin-coating the sample with 300 nm of electro-sensitive resist (ZEP520A), we expose pillars with nominal diameters ranging from 140 nm to 260 nm, in steps of 20 nm. After development, the pattern is transferred into the $SiN_x$ layer by a ICP reactive ion etching process in a F-based plasma ($SF_6$:Ar, 30:15 sccm, ICP power = 500 W, RF power = 35 W, $P$ = 1 Pa) and then etched into the diamond during an ICP process in $O_2$ plasma (99 sccm, ICP power = 1000 W, RF power = 200 W, $P$ = 1 Pa). The final height of the nanopillars is ~500 nm. The remaining nitride layer is finally removed in a solution of buffered HF and the diamond surface is exposed. Emitters investigated in this work are embedded in nanopillars with 180 and 200 nm diameters.

## Optical setup

**Pulse carver.** A Ti:Sa pump laser (Coherent Ultra II) generates the pulses (wavelength: 800 nm, pulse duration: 150 fs, repetition rate: 80 MHz). The clock signal provided from this laser is used to synchronize the rest of the experimental devices.

A commercial frequency conversion setup (APE OPO-X) uses first an optical parametric oscillation process to change the laser wavelength to 1238 nm and then frequency doubling it with a second harmonic generation process to reach 619 nm.

The beam is then directed through a pulse picker (APE Pulse Select) which uses an acousto-optical modulator (AOM) that refracts the selected pulses to a second path. This device is employed by selecting a division ratio such that the repetition rate of the second path is selected as desired.

Next, the beam is inserted into the "pulse carver" setup that is a modified pulse slicer (APE f50). The input beam is first expanded using a telescope for maximal efficiency from the optical components. The pulses reflect off a diffraction grating on which different spectral components are reflected with distinct angles. Then the beam is collimated onto a reflection SLM. Each spectral band impinges on a different position on the SLM. The original design of the pulse slicer had a mirror and slit combination on this position instead.

The SLM can rotate the polarization of the light on each of its pixels by applying voltages. We define a digital slit by selecting pixels that are illuminated by the desired frequency band. A quarter-wave plate in front of the SLM is used such that the pulses maintain a circular polarization before impinging on the SLM to compensate for a possible misalignment of the SLM axis to the original linear polarization of the pulse. The beam is reflected from the SLM with a slight angle such that the combined beam at the diffraction grating can reflect from a mirror below the input beam. A Glan-Taylor polarizer filters the undesired bands out and the tailored pulse exits the setup.

Three more diagnostic paths are prepared at the pulse carver output. The first two are arranged with beamsplitters such that pulses are monitored by a powermeter and a spectrometer throughout the experiments. The third path leads to an autocorrelator for pulse duration measurements (APE Pulse Check NX150) with a flip mirror.

Finally, the main beam is directed into a home-built scanning confocal microscope by a dichroic mirror. An illustration of the pulse preparation setup is provided in Supplementary Fig. 1.

**Confocal microscope.** The excitation mode is directed to the sample in a closed-cycle helium cryostat (Montana S50) at 4.5 K via a home-built confocal scanning microscopy setup with an objective (Zeiss) of 0.9 numerical aperture. A polarizing beam splitter (PBS) is employed to split the excitation and collection paths. A half-wave plate before the PBS is employed to maximize the input power into the cryostat. Another half-wave plate after the PBS is used to optimize the laser polarization for the SnV transition dipole moment.

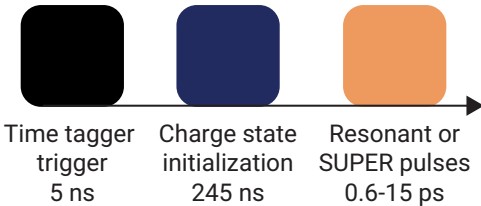

**Fig. 5 | Pulse block diagram for experiments conducted under zero magnetic field.** A trigger signal sent to the time tagger starts the histogram of received counts for the whole sequence. A charge state initialization pulse (blue, 445 nm) ensures the SnV is in its bright state. Finally, a resonant ultrafast pulse or a red detuned pulse pair (for SUPER scheme) excites the emitter, and the subsequent fluorescence signal is measured.

A long-pass filter at 635 nm at the collection path is used to filter the laser light during resonant excitation, and the phonon sideband is coupled to the detection fiber. In photoluminescence (PL) measurements, a long-pass filter at 600 nm is used instead. In the detection fiber the signal is split through a 50:50 fiber beam-splitter, where one port is used to monitor the signal counts (Excelitas SPCM-AQRH-6X-FC-W6), and the other port is either directed to a second channel on the APD for autocorrelation measurements, or to the spectrometer (Andor SR-500i-C-SIL and iDus camera) for PL spectrum measurements. Experiments are controlled via the Qudi[54] software suite.

In addition to the pulsed laser, a blue laser at 445 nm (Hübner Cobolt 06-MLD) for charge state initialization, a green non-resonant excitation laser at 520 nm (DLnSec), and a resonant orange laser at 619 nm (Toptica DL-SHG Pro) for linewidth measurements and spin state initialization/readout are used in the experiments.

**Pulsed measurement scheme.** Our pulsed measurement protocol involves repeating the measurement for a pre-selected integration time and histogramming the arrival times of the photons with respect to a start measurement signal. With the current configuration of the presented experiments, a clock signal from the Chameleon pump laser synchronizes itself with the RF driver module of the Pulse Select. This driver provides a trigger output that we supply to our digital pattern generator (Swabian Instruments Pulse Streamer 8/2).

We program the Pulse Streamer such that after a trigger signal is received, it sends a "Start Measurement" signal to our time tagger (qutools quTAG). After that signal is received, the time tagger starts histogramming the arrival times of the photons until the next start signal resets the clock. Pulse Streamer also sends another trigger signal to our blue laser such that it arrives before the control pulses and the charge state of the emitter is initialized[55]. We set a 245 ns long square blue pulse of ~10 μW continuous-wave power that arrives 672.5 ns before our control pulse (resonant or SUPER excitation) for charge state initialization[55]. The utilized pulse sequence is illustrated in Fig. 5.

**Spin initialization scheme.** We start by conducting photoluminescence excitation spectroscopy and identify the frequencies of the two spin-conserving transitions $\omega_\downarrow$, $\omega_\uparrow$. Then, we stabilize the laser frequency to the lower energetic $\omega_\downarrow$ transition using a PID feedback loop governed by the wavemeter (High Finesse W7). We use an acousto-optical modulator (AA Opto-electronic MT350-A0, 12-VIS modulator and MODA350-B251k-344555 driver) to switch on the laser and prepare square pulses.

The experimental sequence starts with a charge state initialization using a blue 445 nm pulse, and continues with a spin $T_1$ measurement involving initialization/readout pulses (resonant with $\omega_\downarrow$) and a variable delay in between. We insert the SUPER pulses at the beginning of the delay, depending on the chosen configuration. The pulse sequence is presented in Fig. 6.

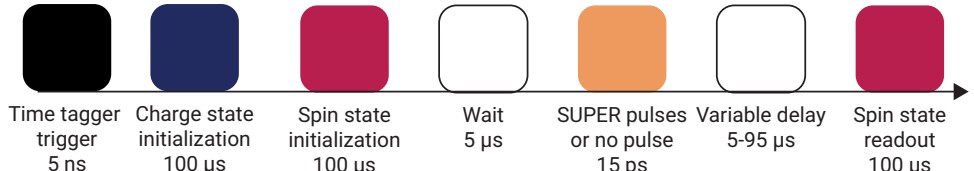

**Fig. 6 | Pulse block diagram for experiments conducted with spin levels.** Under an applied magnetic field, two spin conserving transitions of the SnV become accessible. By tuning a CW laser to one of the transitions, and switching the laser on and off with AOMs it is possible to pump the population between the spin levels (initialize) and read how much of the population is thermalized back to the original state (readout). We histogram the detected photons using a time tagger in repeated measurements. The sequence starts with a charge state initialization pulse (blue, 445 nm) and continues with a spin initialization. After a variable delay, spin state is read out. Adding a SUPER pulse at the beginning of the delay and comparing the thermalization rates allows for probing additional spin mixing effects.

## Simulations

**4-level modeling.** We consider the SnV without a magnetic field as a four-level system as described in the main part of the manuscript, consisting of the states $|1\rangle$ and $|2\rangle$ in the ground-state manifold and states $|3\rangle$ and $|4\rangle$ in the excited state manifold[30]. The Hamiltonian associated with this system reads

$$H_0 = -\frac{\Delta_g}{2}|1\rangle\langle1| + \frac{\Delta_g}{2}|2\rangle\langle2| + \left(\Delta_{ZPL} - \frac{\Delta_u}{2}\right)|3\rangle\langle3| + \left(\Delta_{ZPL} + \frac{\Delta_u}{2}\right)|4\rangle\langle4|$$

$$(1)$$

In this equation, the splitting of the ground states is $\Delta_g = 3.412$ meV, equaling 0.825 THz, while the splitting of the excited states amounts to $\Delta_u = 12.03$ meV (2.91 THz). The total energy separation of ground state to zero-phonon line is 2.003 eV (484.32 THz), but this value does not enter the simulations after a reference frame rotating with this frequency is introduced. The interaction of the defect center with the electric field $\boldsymbol{E}(t)$ of the laser is described in dipole- and rotating wave approximation using the Hamiltonian

$$H_1 = -\frac{\hbar}{2}\Omega(t)\boldsymbol{e}\cdot\boldsymbol{d} + h.c.,$$

$$(2)$$

where $\Omega(t) = \Omega_0(t)e^{-i\omega_L t}$ is the time-dependent laser field strength with frequency $\omega_L$, $\boldsymbol{e}$ is the polarization vector of the electric field and $\boldsymbol{d}$ the dipole vector of the SnV, given by $\boldsymbol{d} = (d_x, d_y, d_z)$. The components of $\boldsymbol{d}$ are given in the basis of the system states $\{|1\rangle, |2\rangle, |3\rangle, |4\rangle\}$,

$$d_x = \begin{pmatrix} 0 & 0 & -0.38 & -0.92 \\ 0 & 0 & -0.92 & 0.38 \\ -0.38 & -0.92 & 0 & 0 \\ -0.92 & 0.38 & 0 & 0 \end{pmatrix}, d_y$$

$$= \begin{pmatrix} 0 & 0 & 0.23i & 0.97i \\ 0 & 0 & -0.97i & 0.23i \\ -0.23i & 0.97i & 0 & 0 \\ -0.97i & -0.23i & 0 & 0 \end{pmatrix}, d_z \qquad (3)$$

$$= \begin{pmatrix} 0 & 0 & 1.95 & -0.47 \\ 0 & 0 & 0.47 & 1.95 \\ 1.95 & 0.47 & 0 & 0 \\ -0.47 & 1.95 & 0 & 0 \end{pmatrix}.$$

We then derive the equations of motion from the Hamiltonian $H_0 + H_1$ using the von-Neumann equation, which is then numerically integrated to calculate the occupation of the system states. Additionally, we use a transformation connecting the SnV frame to the lattice frame, given by $\omega_{SnV} = M \cdot \nu_{lattice}$. Here, $\nu_{lattice}$ is a vector in the lab coordinates, which is transformed to the SnV frame coordinates using

$$M = \begin{pmatrix} 1/\sqrt{6} & 1/\sqrt{2} & 1/\sqrt{3} \\ -2/\sqrt{6} & 0 & 1/\sqrt{3} \\ 1/\sqrt{6} & -1/\sqrt{2} & 1/\sqrt{3} \end{pmatrix}. \qquad (4)$$

Starting with a polarization vector $\boldsymbol{e}_{lab}$ in the lattice frame, this is transformed to the SnV frame before continuing with the evaluation of $\boldsymbol{e} \cdot \boldsymbol{d}$ in Eq. (2). For the laser field, we use a Gaussian envelope reading

$$\Omega_0(t) = \frac{\overline{\alpha}}{\sqrt{2\pi}\sigma}e^{-\frac{t^2}{2\sigma^2}}. \qquad (5)$$

In this formula, $\sigma$ defines the duration of the pulse and is connected to the FWHM of the intensity profile that is given in the main part of the manuscript by $\tau_{FWHM} = 2\sqrt{ln(2)}\sigma$. The amplitude of the laser is defined by $\overline{\alpha}$, however, this is not directly equal to the pulse area $\alpha$. Due to the different coupling strengths for different polarizations of the laser (stemming from the dipole vector), the pulse amplitude connected to the pulse area of $\alpha = \pi$, corresponding to a full inversion of the system under resonant excitation, also depends on the laser polarization. For all calculations, we use a laser polarized in the $x$-direction, i.e., $\boldsymbol{e}_{lab} = (1, 0, 0)$ and scale the pulse area such that under resonance, full inversion is reached for $\alpha = (2n + 1)\pi$ like known from Rabi oscillations. For the SUPER scheme, two pulses are necessary, meaning the expression for the field used in Eq. (2) becomes

$$\Omega(t) = \Omega_1(t)e^{-i\omega_1 t} + \Omega_2(t)e^{-i(\omega_2 t - \varphi)}., \qquad (6)$$

where $\varphi$ is a relative phase between the two pulses. It has been shown that the impact of this phase is negligible unless the pulses are really short, so we neglect it in the calculations. For further details see ref. 27. In the simulations for the two-dimensional map in Fig. 3b of the main manuscript, we fix the less-detuned pulse for the SUPER scheme to a detuning of −0.49 meV to the $|1\rangle \rightarrow |3\rangle$ transition and use a pulse duration of $\sigma_1 = 9$ ps (corresponding to a FWHM of the intensity profile of about 15 ps) together with a pulse area of $\alpha_1 = 6.5\pi$. The second pulse uses the same duration of $\sigma = 9$ ps.

**8-level modeling.** In the presence of a static magnetic field the SnV is modeled by including the Zeeman interaction term[49,56]. The unperturbed basis states are given by

$$|e_{bx}, \downarrow\rangle, |e_{bx}, \uparrow\rangle, |e_{by}, \downarrow\rangle, |e_{by}, \uparrow\rangle, \qquad (7)$$

where $b \in \{g, u\}$ denotes the ground ($g$) or excited ($u$) state manifold. The full system Hamiltonian is

$$H_{G4V} = \begin{pmatrix} H^g & \mathbf{0} \\ \mathbf{0} & H^u \end{pmatrix}. \qquad (8)$$

**Table 1 | Parameters for the Hamiltonian $H_{SnV}$[30]**

| b | $\delta^b$/THz | $\lambda^b$/GHz | $\Upsilon_x^b$/GHz | $\Upsilon_y^b$/GHz | $q^b$ |
|---|---|---|---|---|---|
| g | 0 | 407.5 | 65 | 0 | 0.15 |
| u | 484.34 | 1177.5 | 855 | 0 | 0.15 |

The Hamiltonians of the ground- and excited-state manifolds are given by

$$H^b = H_A^b + H_{SO}^b + H_{JT}^b + H_{ST}^b + H_Z^b ,\tag{9}$$

with

$$H_A^b = \delta^b \mathbb{1},\tag{10}$$

$$H_{SO}^b = \begin{pmatrix} 0 & 0 & -i\lambda^b & 0 \\ 0 & 0 & 0 & i\lambda^b \\ i\lambda^b & 0 & 0 & 0 \\ 0 & -i\lambda^b & 0 & 0 \end{pmatrix},\tag{11}$$

$$H_{JT}^b = \begin{pmatrix} \Upsilon_x^b & 0 & \Upsilon_y^b & 0 \\ 0 & \Upsilon_x^b & 0 & \Upsilon_y^b \\ \Upsilon_y^b & 0 & -\Upsilon_x^b & 0 \\ 0 & \Upsilon_y^b & 0 & -\Upsilon_x^b \end{pmatrix},\tag{12}$$

$$
\begin{aligned}
H_Z^b = \quad & q^b \gamma_L \begin{pmatrix} 0 & 0 & iB_z & 0 \\ 0 & 0 & 0 & iB_z \\ -iB_z & 0 & 0 & 0 \\ 0 & -iB_z & 0 & 0 \end{pmatrix} \\
& + \gamma_S \begin{pmatrix} B_z & B_x - iB_y & 0 & 0 \\ B_x + iB_y & -B_z & 0 & 0 \\ 0 & 0 & B_z & B_x - iB_y \\ 0 & 0 & B_x + iB_y & -B_z \end{pmatrix}
\end{aligned}\tag{13}
$$

Here, $H_A^b$ denotes the unperturbed Hamiltonian, $H_{SO}^b$ the spin–orbit interaction with coupling strength $\lambda^b$, $H_{JT}^b$ the Jahn–Teller contribution with parameters $\Upsilon_x^b$ and $\Upsilon_y^b$, and $H_Z^b$ the Zeeman term lifting the spin degeneracy. The magnetic field is $\boldsymbol{B} = (B_x, B_y, B_z)$, where $\gamma_L = e/2m_e$ and $\gamma_S = \gamma_L$, with $e$ the elementary charge and $m_e$ the electron mass; $q^b$ denotes the orbital reduction factor. The parameters for the SnV are listed in Table 1.

The interaction of a G4V with a classical electromagnetic field is described by

$$H_{\text{Control}}(t) = -\boldsymbol{\mu} \cdot \boldsymbol{E}(t)\tag{14}$$

where $\boldsymbol{E}(t)$ is the time-dependent electric field and the components of the transition dipole operator $\tilde{\mu}^j = -d_j \otimes \mathbb{1}$ are

$$d_x = ae \begin{pmatrix} 0 & 0 & 1 & 0 \\ 0 & 0 & 0 & -1 \\ 1 & 0 & 0 & 0 \\ 0 & -1 & 0 & 0 \end{pmatrix},\tag{15}$$

$$d_y = ae \begin{pmatrix} 0 & 0 & 0 & -1 \\ 0 & 0 & -1 & 0 \\ 0 & -1 & 0 & 0 \\ -1 & 0 & 0 & 0 \end{pmatrix},\tag{16}$$

$$d_z = 2ae \begin{pmatrix} 0 & 0 & 1 & 0 \\ 0 & 0 & 0 & 1 \\ 1 & 0 & 0 & 0 \\ 0 & 1 & 0 & 0 \end{pmatrix},\tag{17}$$

The scaling factor $a$ is determined from the measured excited-state lifetime using Fermi's golden rule $a = 55$ pm. For a static magnetic field that is aligned with the symmetry axis, the Hamiltonian remains Block-Diagonal with respect to the spin degree of freedom. For such a choice of field, diagonalization of the orbital degree of freedom leads us to introduce the new basis states for the ground state manifold $\{|1, \downarrow\rangle, |1, \uparrow\rangle, |2, \downarrow\rangle, |2, \uparrow\rangle\}$ and for the excited state manifold $\{|3, \downarrow\rangle, |3, \uparrow\rangle, |4, \downarrow\rangle, |4, \uparrow\rangle\}$. For the spin-preserving transition of $\frac{1}{\sqrt{2}}(|1, \downarrow\rangle + |1, \uparrow\rangle) \rightarrow \frac{1}{\sqrt{2}}(|3, \downarrow\rangle + |3, \uparrow\rangle)$ using SUPER as shown in Fig. 4a, b, we use two pulses with $\sigma = 3$ ps with pulse areas of $\bar{\alpha}_1 = 9\pi, \bar{\alpha}_2 = 6.21\pi$ and detunings of $\Delta_1 = -5$ meV, $\Delta_2 = -13.0$ meV, both with respect to the $|1, \downarrow\rangle \rightarrow |3, \downarrow\rangle$ transition.

## Data availability

The data and code that support the findings of this study have been deposited in the Zenodo repository with https://doi.org/10.5281/zenodo.17949333[57].

## Code availability

The simulation code that support the findings of this study have been deposited in the Zenodo repository with https://doi.org/10.5281/zenodo.18428586[58].

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

## Acknowledgements

The authors would like to thank Kilian Unterguggenberger for his support when implementing the measurement automation codes; Fransizka Marie Herrmann for her help with analyzing the quasi-continuous Rabi oscillations; Daniel Vajner for sharing his insights regarding the analysis of pulsed Rabi oscillations; Alejandro Schössel for his insights about pulse autocorrelation analysis; Yusuf Karlı, Laura Orphal-Kobin and Enrico Michael for their support regarding the literature review; Alim Yolalmaz and Berk Özkaralı for sharing their knowledge regarding the SLMs; Adrian Runge, Dominik Sudau and Fred Wustmann for performing the SiNx and Ti depositions; Alex Kühlberg and Andreas Thies for

performing the ion implantation; Ina Ostermay, Ralph-Stephan Unger and Olaf Krüger for fruitful discussions; Christoph Becher for his support with organizing the high-pressure high-temperature annealing; Edlef Büttner and Tobias Grunske for their support when modifying the APE Pulse Slicer; and the HOLOEYE Photonics AG for providing the SLM in the pulse carver setup. The authors acknowledge funding by the European Research Council (ERC, Starting Grant project QUREP, No. 851810, awarded to T.S.) and the German Federal Ministry of Education and Research (project QPIS, No. 16KISQ032K, awarded to T.S and G.P.; project QPIC-1, No. 13N15858, awarded to T.S.; project QR.X, No. KIS6QK4001, awarded to T.S.). M.G. was supported by the Turkish Ministry of National Education through the YLSY Scholarship Program.

## Author contributions

M.G., M.I.M., C.G.T. and J.H.D.M. designed and constructed the pulse carver setup. M.G. and C.G.T., developed the experimental protocol, conducted measurements, and analyzed experimental data. S.B. conducted emitter characterization measurements and analyzed experimental data. Ö.O.N. developed experimental software. T.K.B., G.P. and D.E.R. developed and conducted the simulations. M.E.S., D.B.A, M.L.M., and T.P. processed the sample and fabricated nanopillars. T.S. and D.E.R. developed the project. C.G.T., M.G., T.K.B., G.P., D.E.R. and T.S. wrote the paper with input from all authors.

## Funding

## Competing interests

The authors declare no competing interests.
