## [Transparent Peer Review file · Nature Communications]

SUPER and femtosecond spin-conserving coherent excitation of a tin-vacancy color center in diamond

Corresponding Author: Professor Tim Schröder

Version 0:

Reviewer comments:

Reviewer #1

(Remarks to the Author)

The authors demonstrate coherent optical excitation of a tin-vacancy center in diamond using ultrashort off-resonant laser pulses as proposed in the SUPER scheme. Compared to resonant excitation, off-resonant excitation enables separation of excitation and signal via spectral filtering, which otherwise is achieved through complex methods like cross-polarization or filtering in the time or spatial domain. They also demonstrate optical control using both resonant narrowband picosecond and broadband femtosecond laser pulses. Furthermore, they propose a spin-spin entanglement protocol and, through simulations, show that equal spin superpositions can be achieved in the excited state even when the initial state is an unequal superposition. This is important because it relaxes the requirements on precise initial state preparation and scenarios where magnetic field alignment to the SnV quantization axis is not possible, making the protocol practical for a wider range of experimental conditions.

Overall, the paper presents interesting and advanced work, and I believe it is well-suited for publication in Nature Communications once the following points have been addressed.

I have some major remarks on the Results part:

1. The authors report an optical qubit lifetime of 16.2 ns (Fig. 1a), roughly three times longer than expected for SnV centers in bulk diamond. Since they present a very long pure dephasing time of 10.9 ns (Fig. 2b), it would be interesting to discuss the origin of the extended lifetime. Is this a common occurrence in the nanopillar samples due to the proximity to the surrounding surface, or is the observed defect an outlier? In addition, it would be helpful if the utilized CW excitation power could be specified and, if available, the saturation power for resonant CW excitation.
2. Femtosecond coherent excitation regime: The data in Fig. 2e are fit "to a superimposed bi-sinusoidal model, as a second coupled emitter can explain the observed trends in the data". If my understanding is correct, the authors attribute the bi-sinusoidal behavior to coherent excitation of a second SnV center within the broad spectral width of the fs-pulse of ~1400 GHz. However, in Fig. 2f they measure a $g_2(0) \sim 0.2$ (for a π -pulse) which shows no clear evidence of a second emitter, contradicting the previous statement. Is a g_2 measurement available for a 3π -pulse excitation (corresponding to a $\sim 4\sqrt{pJ}$ pulse energy)? If yes, the g_2 should be increased in the presence of a second emitter (which should roughly see a π -pulse at the mentioned pulse energy). While I understand that the bandwidth is chosen to avoid excitation of the D transition, is this still the case at higher pulse energies? Do nonlinear effects play a role?
3. In the simulation of the spin state properties under SUPER control the authors refer to the full SnV Hamiltonian in Ref. [48], where the spin-Zeeman term is incorrect by a factor of two. In the Methods section, describing the 8-level modeling, the spin-Zeeman interaction is noted correctly (although without mentioning orbital quenching). Precise population transfer into the optical excited state is a critical component of the proposed spin-spin entanglement protocol, particularly for magnetic fields misaligned with the SnV high-symmetry axis. Here "preparing an unequal superposition prior to the excitation" would be required to obtain an equal superposition after SUPER excitation, as stated by the authors. Since the spin eigenstates mix proportionally to the spin-Zeeman interaction/ B_{\perp} , could the authors clarify whether the correct gyromagnetic ratio for the spin-Zeeman term was used in their simulations? If not, the simulations in the supplementary Figure 10 should be corrected.
4. If available, including PL and PLE measurements of the investigated SnV center in the supplementary information would help to clarify the previous remarks.

Finally, I have two minor remarks:

5. SUPER scheme: In the main text “detuning” refers to the offset from the central emission frequency for both pulses. However, in Fig. 3a the detuning axis could also be interpreted as the offset between the two pulses, since the blue arrow is pointing toward the pictogram with the diverging pulses. Given the novelty of the SUPER scheme in the community, I suggest clarifying this either in the figure caption or by adapting the pictogram to avoid misconception.

6. In Fig 4c it would be helpful to indicate the CW power used during the initialization pulse, as initialization saturates at a relatively high value compared to the dark background. Could the authors comment on the possible reasons?

Reviewer #2

(Remarks to the Author)

full report available as attached pdf file

Reviewer #3

(Remarks to the Author)

Version 1:

Reviewer comments:

Reviewer #1

(Remarks to the Author)

The authors have made proper modifications, and I believe that the manuscript can be accepted for publication in Nature Communication in the current form.

Small note: on page 4 under Optical Qubit Coherence, “..a single transition optically transition..” has an extra “transition” and the word order is off.

(Remarks on code availability)

Reviewer #2

(Remarks to the Author)

I want to thank the authors for the care, diligence and effort they put in revising the manuscript, thus strengthening their contents as well as offering additional insight to less specialized readers.

The remarks and concerns I raised were addressed satisfactorily and honestly, and I appreciated the additional insights and clarifications given to my questions.

As a consequence, I recommend the paper for publication in its current form

(Remarks on code availability)

Reviewer #3

(Remarks to the Author)

(Remarks on code availability)

Reviewer 1

The authors demonstrate coherent optical excitation of a tin-vacancy center in diamond using ultrashort off-resonant laser pulses as proposed in the SUPER scheme. Compared to resonant excitation, off-resonant excitation enables separation of excitation and signal via spectral filtering, which otherwise is achieved through complex methods like cross-polarization or filtering in the time or spatial domain. They also demonstrate optical control using both resonant narrowband picosecond and broadband femtosecond laser pulses. Furthermore, they propose a spin-spin entanglement protocol and, through simulations, show that equal spin superpositions can be achieved in the excited state even when the initial state is an unequal superposition. This is important because it relaxes the requirements on precise initial state preparation and scenarios where magnetic field alignment to the SnV quantization axis is not possible, making the protocol practical for a wider range of experimental conditions.

Overall, the paper presents interesting and advanced work, and I believe it is well-suited for publication in *Nature Communications* once the following points have been addressed.

We thank the reviewer for their positive comments and recommending our work for publication in *Nature Communications*.

I have some major remarks on the Results part:

1. The authors report an optical qubit lifetime of 16.2 ns (Fig. 1a), roughly three times longer than expected for SnV centers in bulk diamond. Since they present a very long pure dephasing time of 10.9 ns (Fig. 2b), it would be interesting to discuss the origin of the extended lifetime. Is this a common occurrence in the nanopillar samples due to the proximity to the surrounding surface, or is the observed defect an outlier? In addition, it would be helpful if the utilized CW excitation power could be specified and, if available, the saturation power for resonant CW excitation.

The reviewer is correct on pointing out the longer than usual lifetime. For further clarification, we have added an additional paragraph explaining this phenomenon. Indeed, lifetimes reaching up to 10 ns are observable with emitters embedded in nanopillars, attributed to a lower effective refractive index compared to the bulk material. We explain the additional lifetime extension with the polarization based configuration of our confocal microscope. We overlap the collection and excitation paths in our microscope using a polarization beamsplitter (PBS). After the PBS, the SnV is excited with polarization optimized for the C transition; however, photons emitted from this transition are not directed to the collection port. Therefore, the collected photons are mostly emitted via the D transition (phonon sideband). Photoluminescence spectroscopic results typically show 20-33% of the emission occurs from this channel (comparing the peak heights). Therefore, one can estimate the lifetime restricted to this transition could be measured around 16 ns.

We have included the resonant power in the main text and added a saturation measurement to the Supplementary Information following the reviewer's suggestion.

2. Femtosecond coherent excitation regime: The data in Fig. 2e are fit "to a superimposed bi-sinusoidal model, as a second coupled emitter can explain the observed trends in the data". If my understanding is correct, the authors attribute the bi-sinusoidal behavior to coherent excitation of a second SnV center within the broad spectral width of the fs-pulse of ~ 1400 GHz. However, in Fig. 2f they measure a $g_2(0) \sim 0.2$ (for a π -pulse) which shows no clear evidence of a second emitter, contradicting the

previous statement. Is a g^2 measurement available for a 3π -pulse excitation (corresponding to a $\sim 4\sqrt{pJ}$ pulse energy)? If yes, the g^2 should be increased in the presence of a second emitter (which should roughly see a π -pulse at the mentioned pulse energy). While I understand that the bandwidth is chosen to avoid excitation of the D transition, is this still the case at higher pulse energies? Do nonlinear effects play a role?

We thank the reviewer for their suggestion. We have a recorded 3π g^2 measurement from right after the presented Rabi measurement. The data are plotted as follows:

In the time of our original submission, we have not included this plot as we had not fully understood the profile of the presented g^2 pattern. Integrating the $+125$ ns area and taking the ratio with 125 - 375 ns area yielded $g^2(0) = 0.9$.

However, upon further analysis we now believe the shape is generated due to secondary excitations from the imperfectly suppressed pulses at the pulse train (see also Supplementary Figure 7):

We would like to note that, we have verified the countrates presented in Fig 2.c,e are not influenced by the subsequent pulses by carefully arranging the integration window limits. This can be further verified by the close to 0 value acquired at 5π pulse energy ($\sim 7\sqrt{pJ}$).

However, since the central dip at the g^2 measurement is significantly close to the zero value, we agree with the reviewer that the hypothesis of a secondary emitter seems unlikely. For this reason, we have now added to the text that the reason for the varying rotation visibility of the Rabi rotations could be caused by uncharacterized nonlinear effects or detuned coupling to other transitions.

3. In the simulation of the spin state properties under SUPER control the authors refer to the full SnV Hamiltonian in Ref. [48], where the spin-Zeeman term is incorrect by a factor of two. In the Methods section, describing the 8-level modeling, the spin-Zeeman interaction is noted correctly (although without mentioning orbital quenching). Precise population transfer into the optical excited state is a critical component of the proposed spin-spin entanglement protocol, particularly for magnetic fields misaligned with the SnV high-symmetry axis. Here “preparing an unequal superposition prior to the excitation” would be required to obtain an equal superposition after SUPER excitation, as stated by the authors. Since the spin eigenstates mix proportionally to the spin-Zeeman interaction/ B_{perp} , could the authors clarify whether the correct gyromagnetic ratio for the spin-Zeeman term was used in their simulations? If not, the simulations in the supplementary Figure 10 should be corrected.

We thank the reviewer for their comment and correctly placed concern. The simulations were conducted with the correct coefficient. We have now noted the correction in the methods section.

4. If available, including PL and PLE measurements of the investigated SnV center in the supplementary information would help to clarify the previous remarks.

We thank the reviewer for their suggestion. The spectroscopic data is now included in the supplementary information. We would like to note that the PL spectrum was acquired with the PBS collection configuration under an arbitrary polarization. Therefore, while the peak positions are reliable, their relative amplitudes are modified by the polarization filtering.

Finally, I have two minor remarks:

5. SUPER scheme: In the main text “detuning” refers to the offset from the central emission frequency for both pulses. However, in Fig. 3a the detuning axis could also be interpreted as the offset between the two pulses, since the blue arrow is pointing toward the pictogram with the diverging pulses. Given the novelty of the SUPER scheme in the community, I suggest clarifying this either in the figure caption or by adapting the pictogram to avoid misconception.

We thank the reviewer for this comment. We have updated the pulse pair illustration and caption of Fig 3a to clarify the definition of the axes.

6. In Fig 4c it would be helpful to indicate the CW power used during the initialization pulse, as initialization saturates at a relatively high value compared to the dark background. Could the authors comment on the possible reasons?

We have utilized 1 uW of resonant power for spin state initialization and now noted this in the text.

We attribute the non-perfect initialization of the spin states to short T_1 times (~60 us, see Fig 4.d) limited by phononic processes due to the operating temperature of 4.5 K.

Reviewer 2

We thank the reviewer for their positive comments regarding their criteria of validity, significance, methodology, clarity and literature review.

Suggested improvements

0. As a minor comment, I believe that the self-citations [49] and [50] might be unconvincing and weakly linked to the rest of the discussion.

The citations were provided as a part of the outlook section since we would like to combine such excitation schemes with our own ongoing work with integrated chips. However, following reviewer's point that this outlook is not directly linked to the scientific scope of this work, we decided to remove the paragraph.

1. Attribution to SnV center and emission lifetime

- The reader should be convinced that the color center under test for coherent control is actually a SnV center. Any spectral analysis showing the signature of the defect is missing in the main text and, to the best of my understanding, in the supplementary materials.

This cannot be taken for granted.

We thank the reviewer for pointing at the missing SnV confirmation. The spectroscopic data is now included in the supplementary materials, verifying SnV emission by presenting its characteristic spectral peaks at 619.1 and 620.2 nm.

- More of the same, the emitter lifetime (e.g. Fig. 1a) is large with respect to values consolidated in the literature, as also stated explicitly by the authors without further comments. This is where a spectral fingerprint could be a useful proof for SnV attribution. Still on lifetime: multiple pulsed measurements are reported, e.g. Fig. 2d,f; Fig. 3c. Are they all consistent (i.e. excitation-scheme-wise) in the lifetime estimation? This can be commented

We thank the reviewer for suggesting this sanity check analysis. In addition to our explanation about the measured long lifetimes below (see also the comment on this to the other reviewer) and in the manuscript, we have now included the requested data analysis. Fitting the values in the narrowband data between 0.1 and 0.25 $\sqrt{p_j}$ (up to a π pulse energy), we found a lifetime of 16 ns on average with a statistical standard deviation of 2 ns.

- In case SnV is spectrally confirmed and lifetime estimation is so long, can a comment be made? Can this originate from the fact that the sample underwent HPHT annealing? Or to nanopillars fabrication? These effects have been studied, if not for SnV for other group- IV impurities, so a context can be offered.

We have added an additional paragraph explaining this indeed surprisingly long lifetime to the text where the optical lifetime is presented. Indeed, we have observed extended lifetimes reaching up to 10 ns with emitters embedded in nanopillars (not shown). Additionally, due to the polarization based configuration of the confocal microscope, excitation of the SnV is optimized to C transition dipole moment, and the collection path is matching the orthogonal polarization of the photons emitted via the D transition phonon side band. Photoluminescence spectroscopy usually shows 33% of the fluorescence emitted via this optical transition (comparing the peak heights), which in turn can result in measuring an “extended” lifetime around 16 ns when considering $\tau_{\text{lifetime}}=1/(2*\pi*(\Gamma_C*0+\Gamma_D*1))$.

2. Sample preparation

- The authors declare that they performed implantation of both Sn and S ions, but the reason for such choice is not discussed. A reference and a short motivation could be beneficial to the reader.

The sulphur implantation step was included for potential improvement of the emitter yield after the annealing process. This effect has not been systematically characterized with this sample. Its presence, however, does not fundamentally change the SnV properties. However, following the reviewer’s suggestions we provided this information in the methods sections with the suitable reference from Lühmann et al. Nature Communications (2019) 10:4956.

- The manuscript would benefit from the inclusion of material (e.g. SEM pictures, PL scans) of the integrated pillars.

Requested images are added to the supplementary information.

-Spectral analysis could also help clarify whether the entire process resulted in the introduction of strain affecting the behavior of the considered defect. Are the optical lines transform-limited? Was this evaluated during the preliminary characterization and considered as a possible source decoherence?

The inhomogeneous linewidth in many individual PLE scans is indeed much larger than the transform limit (361 MHz > 10 MHz) and therefore could influence the T_2^* measured at Fig 2c. However, since the bandwidths of the pulses used in resonant excitation and SUPER experiments (~40 GHz) are significantly larger than the inhomogeneously broadened line, we believe the interaction strength is relatively robust during repeated and integrated measurements. Moreover, the fact that we determine a coherence time of >10 ns indicates that during the short optical pulses the effective linewidth is only broadened to a small degree.

3. Only concern on statistical significance

- To the best of my understanding this entire work reports data on an individual SnV center embedded in a nanopillar. How reproducible would be the data collected in this work over a population of SnV centers fabricated under the same conditions?

We thank the reviewer for pointing at the generalizability of our measurements. We believe that we can conclude from these data and additional not published data that the methods can be applied over a population of SnV centers fabricated under the same conditions. First, the measurement under the applied magnetic field was conducted on a second, different emitter. This was not referenced, which we have now explicitly stated.

Second, while we have not systematically characterized how a large number of different SnVs respond to the SUPER excitation scheme, we have applied it to a population of three SnVs, which were not pre-selected for particular properties. In addition to the two emitters presented in this paper, we have employed a third emitter (M. Gökçe, Development of Automated and Self-Calibrating Tools for Tin-Vacancy Coherent Control via Spectral Carving of Broadband Pulses, Master Thesis, Karlsruhe Institute of Technology & Humboldt-Universität zu Berlin, 2023, hosted at Humboldt-Universität) again hosted on this sample. All emitters were only preselected on the condition that the nanopillar seemed to host a single emitter. Since a resonance was detected on all of them, although with a limited dataset, we have a strong belief that the measurements are reproducible on other emitters as well

Figure 1 Example for a 3rd emitter to be excited via the SUPER scheme.

Following the reviewer's point, we have now added a comment on this in the manuscript.

- Can the dependence of the coherent drive of the SnV population be considered with respect to the pillar size and shape? Are the emission lines affected by strain or other decoherence sources? (this last comments falls if no strain was detected at the previous point)

The three emitters mentioned above are embedded in nanopillars with nominal diameters of 180 and 200 nm. These nanopillar widths are selected due to the theoretical expectation of having single modes at 619 nm. We have not further investigated the influence of pillar shapes on the emission characteristics.

One of the spectroscopic fingerprints of additional lattice strain on tin-vacancy emitters is an increased ground state splitting compared to the zero strain condition. The measured photoluminescence spectrum shows 819 GHz splitting between the C and D transitions, which is the expected splitting without any additional lattice strain contribution. Therefore, we conclude that the emitter is not under detectable strain.

4. Super SCHEME related comments

- (The present question is relevant also for topics 1-4): SnV center is often reported to suffer from photoblinking. Did the authors observe at any stage (preliminary characterization, conventional spin control, SUPER scheme) or under any specific conditions this effect? Can they comment on the long term photostability of the system and how long the coherent control sequence can be repeated and driven before observing any sort of anomaly?

The reviewer is right that the SnVs in this study are not charge state stable and require periodic reinitialization. For this reason, we apply a charge state initialization pulse at the beginning of each measurement sequence. We have not calibrated the initialization fidelity and therefore do not have a quantitative measure on how many of these sequences the emitter might be in its dark state. However, we have characterized the ionization time under resonant excitation for emitter 1 with 800 nW CW power (~1.5-times the saturation power) and measured it to be 4.5 ms. While the applied powers in pulsed experiments are not directly comparable to the (quasi-)CW measurement, this measurement still provides a valuable indication that the emitter can be photostable for periods much longer than the measurement sequence, which is further demonstrated by the relatively high success rate for a given control sequence.

- Can the authors comment on the challenges to apply the SUPER scheme to different classes of color centers? How parameters must be changed, also in terms of pulse envelopes and detuning? Would this work for emitters with broad emission and absorption band? Would the technique scale to address multiple, spectrally distinguishable SnV emitters or substantial adaptations should be done?

We thank the reviewer for their request for additional information about the general applicability of the SUPER scheme. The following points are now summarized in a paragraph at the SUPER section of the main text.

The main requirement for applying the SUPER scheme is the optical access to a two-level system as its main objective is optical excitation. But of course also atoms and solid state emitters (like the tin-

vacancy) usually contain a more complicated energy manifold with additional levels. So, cross couplings could set limitations on the bandwidth of the two-color pulses and how far you can detune the pulses without addressing other transitions.

Furthermore, if the resonances are broadband due to incoherent effects (such as strong phononic coupling), this could influence the fidelity of the SUPER control. Or, if the broadband nature is a result of very short natural lifetimes, it could set a limit to the SUPER pulse durations as the operation needs to be completed before the excited state relaxes.

The resonance widths of the SUPER scheme are determined by the bandwidths of the utilized pulses. If the emitters are distributed within a 1 GHz spectral width or so, it might not be feasible to individually control each of the emitters in a single diffraction limited spot.

- "Page 5: Experimentally, we find the right configuration of pulses by fixing one pulse at a set frequency, bandwidth, and amplitude; and then performing a scan over detuning and amplitude of the second pulse." Was this achieved by empirical tinkering or followed from specific considerations?

We thank the reviewer about their question and request for the clarification about this point. We have now extended the part in the manuscript on how the resonance conditions are determined.

To clarify the answer in more details: The resonance conditions are not limited to a single combination of two-color pulse parameters. Instead, once of one the pulse parameters are fixed, it sets the conditions on the second pulse. We can theoretically estimate the resonance condition for the second pulse, that is based on the detuning and the pulse strength of the first pulse as shown in previous publications. However, there is still a discrepancy between the pulse area used in theory and in experiments due to the missing knowledge of parameters like the absolute dipole moment strength and the actual strength of the electric field at the position of the defect. Thus, from theory, we estimate the parameter range, in which we expect the second pulse. Roughly speaking it should be slightly above twice the detuning of the first pulse.

The exact resonance is then found by a resonance search scan based on the theoretical considerations, which we would not call empirical tinkering, but rather a theory-guided search.

- Page 4: the population inversion estimate (98%) is large. This should be better clarified or the estimation should be addressed explicitly if made in the supplementary materials. I would relax the statement since "sudden and drastic threshold-based non linear effects" were observed and reported.

Following the reviewer's suggestion, we took the estimated value out of the main text and referred to the supplementary information for the estimation.

- In section Results, subsection Ultrafast resonant coherent control, the authors state that the pulse duration is 50% longer than the expected one due to dispersion introduced on the setup. Did the authors perform an analysis over the different optical components to evaluate such number? It is indeed stated that the authors have different diagnostic paths for the pulse carver but no information is offered regarding the transmission and dispersion. In case additional details are given in the Supplementary information, an explicit statement (e.g., See Supplementary Material) could be given here.

The autocorrelation measurements are conducted on a diagnostic path described in Supplementary Figure 1. We added a comment pointing at this information.

We have not characterized the sources of the dispersion by individually analyzing optical elements. Furthermore, autocorrelation measurements are conducted by bypassing the pulse select for increasing the power. Therefore, measured values do not include additional dispersion that might have been acquired in the pulse select and confocal microscope. We have further clarified this point in the main text, and referenced the supplementary section more explicitly in the main text.

5. Readability

- The arrows included in Fig 3a are of unclear meaning

We have extended the caption of the figure for clarifying the meaning of the axes. We have also inscribed the pulses as 'scan' and 'fixed' for further clarification.

- Fig. 4f might fall out of the scope of the paper and be too tentative (also in terms of actual quantitative information delivered). The manuscript can live without it; the related contents can be presented in a summarized version in the final discussion.

We thank the reviewer for their comment. We would like to emphasize that the question of the usability of the SUPER scheme within entanglement protocols is a very critical question. In particular, this paper for the first time discusses the compatibility of the SUPER scheme (and broadband excitations in general) with spin qubits. By proposing a remote spin-spin entanglement protocol, we provide a very relevant perspective for the aims of the paper. However, following the reviewer's comment we opted to extend the motivation behind proposing such a protocol at the relevant section, and modified Fig 4f (adding the SUPER control fields) to better match this outlook. Furthermore, we have added an additional hint in the Fig. 4f that this the sketch is referring to a proposal.

- Methods section: although the separate supplementary are extensive and complete, I recommend to improve the readability of the integrated Methods section as well. A useful upgrade would be to introduce block diagrams summarizing the schemes adopted for the different pulsed measurements and spin initialization

We thank the reviewer for the comment. The implemented pulse sequences are now illustrated in block diagrams at the methods section.

In this work C. Torun and coworkers discuss an interesting novel approach for the coherent control of the SnV color center in diamond relying on a non-conventional optical scheme with respect to what reported so far in literature, i.e. the Swing-UP of quantum EmitteR population (SWING).

Key results

The authors discuss the utilization of SUPER to achieve optically coherent excitation of the SnV defect spin. The authors use two detuned pulsed lasers to achieve population inversion over the C transition at cryogenic temperature. The diamond sample was fabricated using ion implantation and subsequent nanopillars fabrication to enhance photon collection extraction from the optical defects. The SUPER analysis was preceded by a preliminary characterization of one selected SnV center, aiming at determining T_1 and T_2^* parameters, which were not compatible with values consolidated in the literature. Finally, Rabi oscillations are driven upon different schemes, including the implementation of the SUPER. This latter method is claimed to offer an easier approach from an experimental standpoint to coherent population inversion.

Validity

The driven Rabi oscillation presented in the manuscript are obtained for different techniques: quasi-continuous excitation; narrowband ps pulsed laser, fs pulsed laser; SUPER excitation. The obtained results are consistent and satisfactory in terms of reproducibility over the different techniques. The proposed SUPER scheme is demonstrated to be suitable, providing similar results with respect to the other techniques developed in the scientific literature.

Significance

The experiment presented in this manuscript is well discussed and the data has scientific significance in terms of clarity and methodology. While coherent control has already been demonstrated in the scientific literature, the population inversion fidelity, although estimated (further in the following), is a significant improvement with respect to state of the art. While the results themselves are not necessarily groundbreaking, the main focus of the work is however the demonstration that the SUPER scheme can efficiently drive solid state color centers without the need for a spectrally-invasive resonant excitation. If the applicability is proven for additional spin systems, it could significantly boost the range of application of color centers as it would enable a readout of the ZPL with significantly larger SNR with respect to resonant techniques.

Data and methodology

The data is presented in a satisfactory way. The manuscript is well divided into section allowing the reader to follow the experimental procedures discussed per section. No apparent logical missteps can be spotted. The achievement of compatible results with consolidate techniques offers a clear validation of the SUPER excitation scheme.

The supplementary information is rich of details enabling:

- to implement the scheme at different research laboratories
- to verify the data acquisition and the data fitting models adopted for the analysis

Therefore, the work appears transparent and can be properly falsified in principle by any scientist having access to suitable equipment with a known and clearly presented methodology.

Clarity and context

The manuscript is well written and the sections are clear and filled with pertinent citations and context.

References

The references cited for this manuscript are generally appropriate for such work and provide a comprehensive picture of the general framework of the research field. This holds true for both theoretical/simulation works as well as for the experimental approaches relevant to coherent control of 2 level systems. A clear and up to date overview of the research on diamond color centers is also given.

As a minor comment, I believe that the self-citations [49] and [50] might be unconvincing and weakly linked to the rest of the discussion.

Your expertise

Mostly related to the experimental framework; the evaluation of the theoretical work and simulation discussion shall be regarded as qualitative.

Providing constructive feedback

The manuscript in general is solid. The list of remarks and clarifications in the following aims at improving reproducibility and further improve the work performed by the authors.

As a general remark, reproducibility should not be limited to the high-level implementation of the technique, but it should involve also the necessary steps towards the fabrication of the sample and preliminary assessment of (claimed) SnV centers.

Suggested improvements

There are few improvements to be addressed in order to further consolidate the manuscript to be suitable for publication:

1. Attribution to SnV center and emission lifetime

- The reader should be convinced that the color center under test for coherent control is actually a SnV center. Any spectral analysis showing the signature of the defect is missing in the main text and, to the best of my understanding, in the supplementary materials. This cannot be taken for granted.
- More of the same, the emitter lifetime (e.g. Fig. 1a) is large with respect to values consolidated in the literature, as also stated explicitly by the authors without further comments. This is where a spectral fingerprint could be a useful proof for SnV attribution.
- Still on lifetime: multiple pulsed measurements are reported, e.g. Fig. 2d,f; Fig. 3c. Are they all consistent (i.e. excitation-scheme-wise) in the lifetime estimation? This can be commented
- In case SnV is spectrally confirmed and lifetime estimation is so long, can a comment be made? Can this originate from the fact that the sample underwent HPHT annealing? Or to nanopillars fabrication? These effects have been studied, if not for SnV for other group-IV impurities, so a context can be offered.

2. Sample preparation

- The authors declare that they performed implantation of both Sn and S ions, but the reason for such choice is not discussed. A reference and a short motivation could be beneficial to the reader.
- The manuscript would benefit from the inclusion of material (e.g. SEM pictures, PL scans) of the integrated pillars. Spectral analysis could also help clarify whether the entire process resulted in the introduction of strain affecting the behavior of the considered defect. Are the optical lines transform-limited? Was this evaluated during the preliminary characterization and considered as a possible source decoherence?

3. Only concern on statistical significance

- To the best of my understanding this entire work reports data on an **individual** SnV center embedded in a nanopillar. How reproducible would be the data collected in this work over a population of SnV centers fabricated under the same conditions?
- Can the dependence of the coherent drive of the SnV population be considered with respect to the pillar size and shape? Are the emission lines affected by strain or other decoherence sources? (this last comments falls if no strain was detected at the previous point)

4. Super SCHEME related comments

- (The present question is relevant also for topics 1-4): SnV center is often reported to suffer from photoblinking. Did the authors observe at any stage (preliminary characterization, conventional spin control, SUPER scheme) or under any specific conditions this effect? Can they comment on the long term photostability of the system and how long the coherent control sequence can be repeated and driven before observing any sort of anomaly?
- Can the authors comment on the challenges to apply the SUPER scheme to different classes of color centers? How parameters must be changed, also in terms of pulse envelopes and detuning? Would this work for emitters with broad emission and absorption band? Would the technique scale to address multiple, spectrally distinguishable SnV emitters or substantial adaptations should be done?
- "Page 5: Experimentally, we find the right configuration of pulses by fixing one pulse at a set frequency, bandwidth, and amplitude; and then performing a scan over detuning and amplitude of the second pulse." Was this achieved by empirical tinkering or followed from specific considerations?
- Page 4: the population inversion estimate (98%) is large. This should be better clarified or the estimation should be addressed explicitly if made in the supplementary materials. I would relax the statement since "sudden and drastic threshold-based non linear effects" were observed and reported.
- In section Results, subsection Ultrafast resonant coherent control, the authors state that the pulse duration is 50% longer than the expected one due to dispersion introduced on the setup. Did the authors perform an analysis over the different optical components to evaluate such number? It is indeed stated that the authors have different diagnostic paths for the pulse carver but no information is offered regarding the transmission and dispersion. In case additional details are given in the Supplementary information, an explicit statement (e.g., See Supplementary Material) could be given here.

5. *Readability*

- The arrows included in Fig 3a are of unclear meaning
- Fig. 4f might fall out of the scope of the paper and be too tentative (also in terms of actual quantitative information delivered). The manuscript can live without it; the related contents can be presented in a summarized version in the final discussion.
- Methods section: although the separate supplementary are extensive and complete, I recommend to improve the readability of the integrated Methods section as well. A useful upgrade would be to introduce block diagrams summarizing the schemes adopted for the different pulsed measurements and spin initialization.